# Association of *IL-23R* rs1569922 and Other Probable Frequent Etiological Factors with Legg–Calvé–Perthes Disease in Mexican Patients

**DOI:** 10.3390/genes16101126

**Published:** 2025-09-24

**Authors:** Armando Odiseo Rodríguez-Olivas, Elba Reyes-Maldonado, Leonora Casas-Ávila, Marlene Alejandra Galicia-Alvarado, Erika Rosales-Cruz, Cesar Zavala-Hernández, Edgar Hernández-Zamora

**Affiliations:** 1Genomic Medicine, Instituto Nacional de Rehabilitación “Luis Guillermo Ibarra” (INR-LGII), Mexico City 14389, Mexico; pharmandooro@gmail.com (A.O.R.-O.); lcasasa@gmail.com (L.C.-Á.); 2Morphology Department, Escuela Nacional de Ciencias Biológicas Instituto Politécnico Nacional (ENCB, IPN), Mexico City 07738, Mexico; erika_encb@hotmail.com; 3Neurociencias Clínicas, Instituto Nacional de Rehabilitación “Luis Guillermo Ibarra”, Mexico City 4389, Mexico; marlenegalicia@gmail.com; 4Clinical Pathology Laboratory, Instituto Nacional de Rehabilitación “Luis Guillermo Ibarra”, Mexico City 4389, Mexico; cezaher@yahoo.com.mx

**Keywords:** Legg–Calvé–Perthes disease, etiology, hemostatic, inflammatory, genetics

## Abstract

Background: Legg–Calvé–Perthes disease (LCPD) is a rare avascular osteonecrosis of the proximal femoral epiphysis and typically occurs during the childhood growth phase. LCPD is a complex illness of unknown origin, which is considered the main difficulty in the study of this disease. Various theories on LCPD etiology have been proposed; however, no consensus has been reached about its origin. Our research objective was to evaluate the polymorphisms *FVL* rs6025, *FVIII* rs5987061, *FIX* Malmö rs6048, *PAI-1* rs1799889, *eNOS* rs17899983/rs2070744, *IL-23R* rs1569922/rs154655686/7539625, and *TNF-α* rs180062, and their relationship with LCPD. Methods: A blood sample was taken from each study participant. Complete blood count, coagulation times and factors, antithrombotic proteins, and homocysteine (Hcy) were determined using a coagulometric method. DNA was obtained and genotyped using real-time PCR with TaqMan probes. Genotypic and allelic distributions were analyzed using comparative analysis, the Hardy–Weinberg equilibrium, and OR. Results: This study included 46 children: 23 with LCPD (cases) and 23 without (controls). Statistically significant differences were found in Prothrombin Time, Factor V, and Factor IX activity, as well as Hcy concentration; these values suggest the presence of hypercoagulable states in patients, which can cause thrombotic events. On the other hand, significant differences were also found in the neutrophil–lymphocyte ratio and systemic immune-inflammation index, showing major inflammation states in the patient group. Moreover, statistically significant differences were found in the *IL-23R* rs1569922 polymorphism; it was found that carriers of the T/T and C/T genotypes have an increased risk of developing LCPD. Conclusions: Our results show greater hemostatic activity and inflammation in the group of patients included in this study, supporting various theories previously proposed. Therefore, we believe that LCPD is a multifactorial condition in which hemostatic, inflammatory, and genetic factors play a central and triggering role in the disease.

## 1. Introduction

Rare diseases (RDs) affect more than 300–400 million people worldwide, causing chronic illness, disability, poor quality of life, and premature death. Most RDs affect children and have an underlying genetic cause. However, making a molecular diagnosis is often a challenge [1].

Legg–Calve–Perthes disease (LCPD) is an RD since it has an unknown etiology and low incidence, ranging from 0.4/100,000 to 29.0/100,000 children. Moreover, the illness is five times more common in males than females [2]. It is characterized by avascular necrosis of the head of the femur; in fact, permanent femoral head (FH) deformity is the most significant sequelae, and it can cause deformities in the acetabulum of the hip. Patients report pain during and after physical activity in the affected limb. At first presentation, lameness is observable, and it is common to find limitations of abduction and internal rotation, as well as limitation in flexion of approximately 20 degrees; in some cases, there is shortening of the affected extremity [2,3].

The etiology of LCPD remains unclear despite years of research. Hypercoagulable states, thrombophilic disorders, or hypofibrinolysis were the first conditions identified as having roles in the pathogenesis of this disease. Like the factor V Leiden (*FVL*) mutation, hyperactivity of FVIII, FIX, and prothrombin; alterations in natural anticoagulants such as protein C and S; alterations in plasminogen activator inhibitor-1 (PAI-1) leading to hypofibrinolysis, among other prothrombotic factors [2]. It has been suggested that necrosis of bone tissue is caused by reduced blood supply due to extrinsic compression of vessels or intravascular occlusion—the main causes of decreased blood supply—which results in the collapse of the caput femoris [2,3,4,5].

Structural bone genes have also been associated with the etiology of LCPD, including mutations in the type 1 collagen (COL1A1) and type 2 collagen (COL2A1) genes or polymorphisms [6]. Other studies indicate a pathological repair process of LCPD, which is marked by an imbalance of bone resorption and formation that contributes to the pathogenesis of FH deformity. In this context, cytokines have a central role in maintaining bone homeostasis. Osteoclasts (OCs), sole bone-resorbing cells, are regulated by numerous cytokines, for example, tumor necrosis factor-alpha (TNF-α), interleukin (IL) 1, IL-6, IL-23, and IL-34, which are described as osteoclastogenic cytokines [7,8].

Recently, many studies have reported that IL-23R is related to inflammatory disorders [9]. Moreover, endothelial nitric oxide synthase (eNOS) and TNF-α play a crucial role in the regulation of inflammation, especially in the context of cardiovascular diseases. It has also been proposed that inflammation plays a central role in the etiology of LCPD. This is because, on the one hand, inflammation has an impact on bone modeling. The neutrophil/lymphocyte ratio is a marker of subclinical inflammation, which increases as damage in the FH enlarges. Therefore, it could be considered that inflammation has implications for the appearance of LCPD and its severity.

On the other hand, several environmental factors have been linked to the development of LCPD, such as low birth weight, hyperactivity, and smoking, both during pregnancy and in childhood. That is the current opinion is that this disease is multifactorial and is caused by a combination of metabolic, genetic, and/or environmental factors [2].

The LCPD progresses in four phases: necrosis (death of the FH bone due to lack of blood supply), fragmentation (the dead bone is resorbed and breaks apart), reossification (new bone develops to repair the FH), and remodeling (the FH consolidates and reshapes until the child’s growth is complete) [2]. Treatments focus on preventing or correcting FH deformities and vary considerably [10]. Nonsurgical intervention includes an abduction cast or orthoses for 1.5 to 2 years. This procedure focuses primarily on maintaining the FH within the acetabulum during the remodeling phase. However, it depends on multiple factors, including age at clinical onset, the extent of epiphyseal involvement, the stage of disease, and the degree of FH deformity. Convincing evidence now exists that demonstrates that non-operative treatment is ineffective and should be abandoned [11]. The treatment of LCPD surgical intervention consists of pelvic or femoral osteotomy, which is very successful. However, surgery should be reserved for patients with extensive epiphyseal involvement in late diagnosed and advanced forms of the disease. The prognosis is generally favorable, and patients are expected to exhibit excellent or good results upon follow-up to the age of 40 years [2,12,13].

To establish an appropriate treatment, given the complexity of LCPD and its possible genetic basis, it is necessary to understand its etiology. We have described how various environmental factors can affect the progression of the disease and its treatment. We are beginning to understand the complex processes related to its etiology. The interruption of blood supply to the HF leads to cartilage and bone necrosis, subchondral fracture, compaction of necrotic bone, and revascularization of the necrotic epiphysis from the periphery. Some polymorphisms of the *IL23R* gene are associated with osteonecrosis and may be linked to preexisting pathological inflammatory conditions and immunological aspects, but they have been little studied. Due to these processes, fragile bone with reduced biomechanical properties is produced, which promotes HF deformation. Based on this background, the aim of this study was to evaluate the *FVL* rs6025, *FVIII* rs5987061, *FIX* Malmö rs6048, *PAI-1* rs1799889, *eNOS* rs17899983/rs2070744, *IL-23R* rs1569922/rs154655686/7539625, *TNF-α* rs180062 polymorphisms and their possible relationship with LCPD in a population of Mexican patients.

## 2. Materials and Methods

### 2.1. Patients

This study included 46 children: 23 with LCPD (cases) and 23 healthy individuals (controls). Patients of all ages, whether first-time visitors or recurrent cases, were diagnosed using clinical criteria and radiological signs in the Pediatric Orthopedics Department. The controls were individuals who attended the Pediatric Orthopedics Department with clinical and radiological evidence indicating the absence of abnormalities in the femur and hip. The controls and patients were matched according to age, sex, weight, and body mass index (BMI). All participants agreed to take part in the study, had no thrombotic pathology, and were not receiving pharmacological treatment.

A blood sample was taken from each participant and collected in a tube with EDTA K2 and a tube with 3.8% sodium citrate. All hemolyzed or lipemic samples were discarded. All hemolyzed or lipemic samples were discarded. Blood count was performed in a Coulter LH 780 hematology automated analyzer. Citrated plasma was separated, and the samples were analyzed in a coagulation analyzer IL ACL Elite/Pro using commercial kits (HemosIL™ Werfen, Mexico City, Mexico), for each determination: TT Thrombin Time; PT RecombiPlasTin 2G; APTT-SP (liquid); Factor (F) I; FII; FV; FVII; FVIII; FIX; FX; FXI; FXII; Antigenic von Willebrand factor; Homocysteine; Protein C; Liquid antithrombin.

DNA was extracted from whole blood leukocytes using a commercial kit, Puregene (Qiagen, Mexico City, Mexico), according to the manufacturer’s protocol. SNPs included were *FVL* (rs6025), *FVIII* (rs5987061), *FIX* (rs6048), PAI-1 (rs1799889), *eNOS* (rs17899983, rs2070744), *IL-23R* (rs1569922, rs7518660, rs7539625) *TNF-α* (rs180062), genes were genotyped by real-time PCR, with Taqman^®^ probes labeled with FAM or VIC (Applied Biosystems, Foster City, CA, USA), in a Real-Time Step One PCR System (Applied Biosystems). Real-time PCR re-action mixture was performed in a volume total of 25 mL PCR containing 1X TaqMan PCR master mix, probe at 100 nm, 900 nm of each primer and 25 ng of genomic DNA. Cycling conditions: denaturation at 95 °C for 10 min, followed by 45 cycles at 92 °C for 15 s and then 60 °C for 1 min [2,8].

### 2.2. Statistics

A database in GraphPad Prism version 8.0 for Windows was designed for the data obtained. A normality analysis of the data was performed using the Shapiro–Wilk test, and a comparative analysis was conducted using the Mann–Whitney U test to determine any significant differences between the groups.

The frequencies of genotypes and alleles of the polymorphisms were compared using the chi-square test and the Hardy–Weinberg equilibrium was calculated. The associations between the different variables and the genotypes of the different polymorphisms with risk of LCPD were measured using the Odds Ratio (OR) and the 95% confidence interval (CI) to evaluate relative risk. The associations were considered significant if their level was *p* ≤ 0.05; the statistical analysis was performed using PAST 4.03 and inheritance models for SNPs and associations were estimated using the SNP Stats program (https://www.snpstats.net/start.htm accessed on 26 September 2024). A binary logistic regression analysis was performed using the Wald method (forward stepwise) to identify risk markers related to the presence of LCPD, based on the contributions of the following indicators: PT, FIX, Hcy, and *IL-23R* rs1569922 polymorphism; the degree of fit was estimated using the classification table. Linear correlation values between predictor variables of the model were determined to verify the assumption of collinearity.

### 2.3. Ethical Aspects

The patients included in this study were diagnosed with LCPD through clinical and radiological assessments. The controls were individuals without radiological alterations in the femur and hip and with no history of thrombophilia or any other ailment. Both groups were selected under the guidelines of Norma Oficial Mexicana, NOM-253-SSA1-2012, for blood banks. All participants received oral and written information about this study and signed a letter of consent. This study was carried out in accordance with the Declaration of Helsinki and received approval from the Ethics, Research, and Biosafety Committees of INR-LGII (approval code: INR 54/16; approval date: 15 August 2016).

## 3. Results

A total of 23 patients (21 men and 2 women) and 23 controls (21 men and 2 women) were enlisted and matched by sex and age. The patients had a mean age of 16.8 years and a mean age at diagnosis of 5.2 years; the mean age for the control group was 16.6 years. The average height of the patients was 1.5 ± 0.2 m and 1.5 ± 0.2 m in the controls, while the average weight was 46.2 ± 15.0 kg in the patients and 54.1 ± 23.5 kg in the controls. In addition, BMI was 20.2 in the patients and 23.1 in the controls. No significant differences were found in weight, height and BMI between the two groups.

Some patients were first-time consultations and others recurrent; however, the clinical diagnosis described by the Pediatric Orthopedics Service indicated that all patients presented early stages of necrosis and fragmentation. Figure 1 shows a control individual and a patient with a clinical diagnosis of LCPD. Three patients presented bilateral involvement: in five of them, the affected limb was the right one, and in fifteen, the left one. During clinical interviews, some of the patients described relevant data; for example, most of the patients indicated the practice of some extreme sport, such as gymnastics, taekwondo, or soccer/football. Around 65% of our patients had pes planus, which has been linked to elevated stress loading of the hip. In addition, approximately 91% of our patient population reported having been exposed to maternal prenatal smoking and/or passive exposure to tobacco smoke, since one or both parents, or a close relative were smokers (Figure 2).

In the blood test, hemoglobin, leukocytes, and platelets presented a similar distribution, without significant differences between the patients and the controls (Table 1). Regarding coagulation times, Thrombin Time (TT) and activated Partial Thromboplastin Time (APTT), did not present significant differences between both groups. However, Pro-thrombin Time (PT) was lower in the patients and presented a significant difference. On the other hand, significantly higher means with a medium effect size (greater than 0.3) were found for coagulation FV activity and FIX activity, as well as for homocysteine (Hcy) concentration, while FVIII was close to reaching significance (Figure 3, Table 2). In addition, higher values were found in the patient group for the systemic immune-inflammation index (SII) and the neutrophil–lymphocyte ratio (N/L R) (Figure 4, Table 3).

The genotype and allelic frequency distributions in both the patients and the controls are summarized in Table 4. For all participants, DNA analyses of *FVL*, *FVIII*, *FIX*, *PAI-1*, *eNOS*, *IL-23R* and *TNF-α* were performed. There were no significant differences between the cases and the controls. Significant differences in allele frequencies were detected in the interleukin 23 receptor (*IL-23R*) between the cases of LCPD and the controls. The genotype frequencies of *IL-23R* polymorphism rs1569922 were in the Hardy–Weinberg equilibrium. The variant T allele was present in 95.6% of the cases with LCPD and in 69.5% of the controls. The T allele was associated with an increased risk of developing LCPD. Adjusting the data for the SII and N/L R, it was found in the codominant and recessive models that carriers of the T/T and C/T genotypes have a higher risk of developing LCPD (Table 4).

According to the logistic regression model using the Wald method, the variables selected with statistical significance (*p* < 0.05) were PT and *IL-23R* rs1569922. Therefore, the resulting logistic function was Y = −3.90 + 0.002 (PT)+ 3.31 (*IL-23R* TT) + 2.45 (*IL-23R* CT).

The OR value of the PT variable was 1.00 (95% CI: 1.001–1.003), the OR value of the *IL-23R* TT variable was 27.64 (95% CI: 2.08–367.41), and the CT value for *IL-23R* was 11.64 (95% CI: 1.04–130.32). Therefore, the presence of lower PT values and a higher frequency of *IL-23R* were determined to be factors that showed an association in this study population for LCPD (Figure 5). Cox and Snell’s R square and Nagelkerke’s R square coefficients of determination indicated that 28.9% and 38.5% of the total residual variance was explained by the presence of PT and *IL-23R* rs1569922 variables included in the model.

Likewise, the classification table demonstrated that the percentage of the correct classification for the positive cases was 73.9% (sensitivity of model) and 78.3% for the controls (specificity of model). In total, 35 individuals were correctly classified by the model, which represented a 76.1% overall fit.

## 4. Discussion

The LCPD represents a global health problem, and its etiology is still uncertain. While our objective was to evaluate different polymorphisms and their possible relationship with LCPD, it is interesting to highlight the presence of environmental factors described in the literature, such as mechanical stress associated with extreme sports and exposure to tobacco smoke, present in this population [13].

Hemostatic disorders have been studied as probable etiological factors of LCPD. The presence of Factor V Leiden (FVL) and prothrombin C20210A mutations, deficiencies in proteins C and S, elevated levels of lipoprotein (a) or fibrinogen, as well as FV and FVIII hyperactivity have been reported in populations with LCPD [2,3,4,5,6]. In this study, we found significant differences between the patients with LCPD and the controls in PT, FV, FIX, and Hcy.

PT is a routine study in the clinical laboratory. A lower PT is related to greater hemostatic activity since less time will be required for thrombus formation. Studies have been described in which PT could be useful, together with other hemostatic markers, in the diagnosis of LCPD [14].

Coagulation Factor V is an essential cofactor for FX in the common pathway of coagulation. The increase in activity of FV, such as the one we found in the patients in this study, has been related to thrombosis. On the other hand, several studies have described the high prevalence of the FVL in Caucasian populations. This mutation has been linked to a high risk of venous thromboembolism, deep vein thrombosis (DVT), pulmonary embolism, and thrombosis in unusual locations. However, it has also been reported that this mutation is not present in the Mexican population, and it was not found in the participants of this study [4,5,14]. Increased coagulation Factor IX activity has been linked to DVT, particularly in the upper extremities, and to LCPD [14,15,16,17].

In this study, a significant increase in Hcy levels was observed in the patients with LCPD relative to the controls. An increase in plasma Hcy is associated with an increased risk for DVT, coronary artery disease, and atherosclerosis; moreover, it has also been associated with osteoporosis and LCPD [18,19,20,21].

On the other hand, cigarette smoke exposure, which was present in our patients, seems to affect the hemostatic process via multiple mechanisms, including alterations in the functions of endothelial cells, platelets, fibrinogen, and coagulation factors, consequently exacerbating the risk of thrombophilia [22].

Not only have mechanical and metabolic disorders been linked to LCPD, but also genetic alterations. Polymorphisms and mutations such as *FVL*, *COL2A1*, G-455-A polymorphism of the *β fibrinogen*, *MTHFR*, *eNOS* polymorphisms, *IL-6* polymorphism, and others related to LCPD have been reported in recent decades, showing genetics as a central etiological factor in LCPD [5,6,7,8,21,22,23,24,25,26,27,28,29,30,31].

The *IL-23* gene is in 12q13.3 and its functional receptor (IL-23R) consists of a heterodimer between IL-12Rβ1 and IL-23R. IL-23R is produced by antigen-presenting cells; however, inflammatory macrophages express IL-23R and are activated by IL-23 to produce IL-1, tumor necrosis factor-alpha (TNF-α), and IL-23 itself. IL-23 stimulates a proinflammatory condition, leading to damage in various structures involving the joints, bones, and cartilage. On the other hand, IL-23 is also involved in osteoclast genesis and bone destruction, independently from IL-17, via induction of receptor activator of kappa B ligand (RANKL) expression in T cells and tartrate-resistant acid phosphatase (TRAP) activity of osteoclasts [32,33,34,35,36,37,38]. It has also been described that *IL-23R* rs1569922 polymorphism may play an important role in the development of osteonecrosis of the FH (ONFH), a condition that occurs in young adults whose average age varies, in the literature, between 27 and 36 years [39].

As previously mentioned, inflammation appears to play an important role in the etiology of LCPD. Therefore, studying the clinical significance of the SII in LCDP is relevant. The SII is based on peripheral lymphocyte, neutrophil, and platelet counts and has been considered a good index that reflects the local immune response and systemic inflammation. Furthermore, the SII may provide useful biomarkers for assessing osteopenia/osteoporosis and other diseases [40,41,42,43]. Our results show higher levels in the SII index and N/L R. These results could also be related to the results obtained from the association of IL-23R in the LCPD patients of this study. To date, alterations related to various risk factors associated with LCPD have been described.

The Convergent Coagulation Model was recently reported as a contemporary interpretation of hemostasis that integrates coagulation with immunological processes and innate inflammation in response to vascular alterations [44]. Our results indicate risk markers associated with hypercoagulable states (thrombophilia) and proinflammatory states, markers related to bone structural and metabolic alterations, as well as some environmental factors in the patient group, which support the various theories previously proposed. As described in this and other recent studies, LCPD is a condition of multifactorial origin in which environment, metabolism, and genetics play central and triggering roles (Figure 6) [45].

Molecular patterns such as polymorphisms *IL-23R* rs1569922 in this study are associated with hypercoagulable states (FVL, FVIII, FIX, PAI-1, and MTHFR), inflammatory states (eNOS, IL-6, IL-23, and IL-23R), bone structural alterations (COL1A1, COL2A1), bone metabolism (OPG, RANK, and RANKL), growth factors (platelet-derived growth factor (PDGF) and vascular endothelial growth factor, (VEGF)), and hormonal alterations (Growth Hormone, GH) (Figure 6) [46].

These molecular patterns facilitate diverse interactions within and between systems, not only complementing and reinforcing cellular clot formation but also directing the re-sponse toward clot resolution and wound healing. By extending coagulation beyond its current limits, the convergent model seeks to offer new diagnostics and therapies [47].

Currently, the research on LCPD emphasizes biomarkers for the screening, prognosis, and responses to treatment. One of the recurring recommendations is the need for biomarker standardization and the correlation of biomarkers with clinical and other measurable parameters. Further investigation into the underlying molecular mechanisms of disease pathology would allow the integration of biomarker studies into controlled clinical studies and investigating biological therapies, which will permit significant advances in a field where there are currently many unmet needs. Furthermore, early recognition by primary care providers is crucial for timely referral to orthopedic services, along with interim support through physiotherapy, pain management and access to mental health and educational resources.

The main limitation of this study is its sample size, given that LCPD is a rare disease. Therefore, the results are highly variable, which limits the estimation of predictors of LCPD in this population of individuals. It is worth noting that all our findings are supported by other studies described in different populations of patients with LCPD or in studies related to FH abnormalities. Furthermore, the National Institute of Rehabilitation “Luis Guillermo Ibarra” (INR) is a national and international reference center, and one of the few institutions in Mexico that records functional disability diagnoses and generates statistics aligned with the International Classification of Functioning, Disability, and Health (ICF). However, it would be necessary to attempt to increase the sample size through multicenter studies, thus expanding the participant groups from different and independent populations, with the hope that these variables will continue to be predictors of the disease and, therefore, confirm the results obtained.

Finally, it is important to highlight that the USA, Canada, Japan, England, China, Germany, South Korea, and France are the countries that have published the most in relation to LCPD. Moreover, India, Mexico, and Australia are among the countries that have published more on this topic recently, indicating that LCPD research is gradually receiving attention from countries around the world. However, in Latin America, there is no prevalence of data on LCPD. Currently, Mexico is one of few countries in Latin America conducting research on LCPD [31].

## 5. Conclusions

LCPD is a pathology with an insidious onset that limits timely identification. This study demonstrates the etiological factors present in our population and in other similar studies, such as mechanical overload, hemostatic alterations, inflammation, smoke exposure, etc., as well as for the first time a relationship between *IL-23R* polymorphism rs1569922 and LCPD. And the potential use of biomarkers like PT, Hcy, SII and N/L R and *IL23R* as biomarkers in the diagnosis or monitoring of disease progression. Our data suggests that LCPD is a condition of multifactorial origin in which environment, metabolism, and genetics play central and triggering roles (Figure 7). Overall, as the cause of LCPD remains unclear, and further studies using distinct methods are required to strengthen the research findings of this study.

## Figures and Tables

**Figure 1 genes-16-01126-f001:**
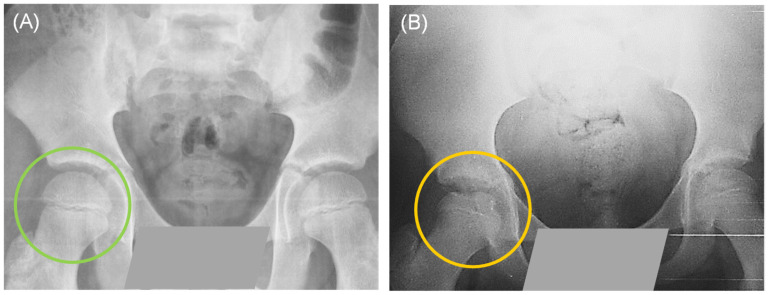
Simple pelvic X-ray in anteroposterior projection: (**A**) The green circle shows the intact femoral head of a healthy control. (**B**) The yellow circle shows a fragmented femoral head of a patient with Legg-Calvé-Perthes disease (LCPD).

**Figure 2 genes-16-01126-f002:**
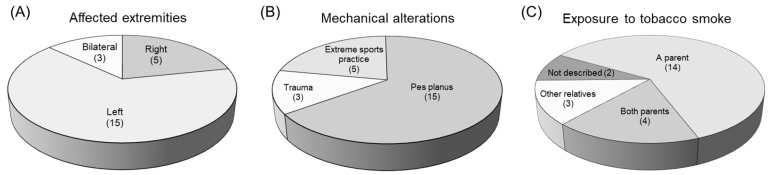
Some relevant clinical data in the patients.

**Figure 3 genes-16-01126-f003:**
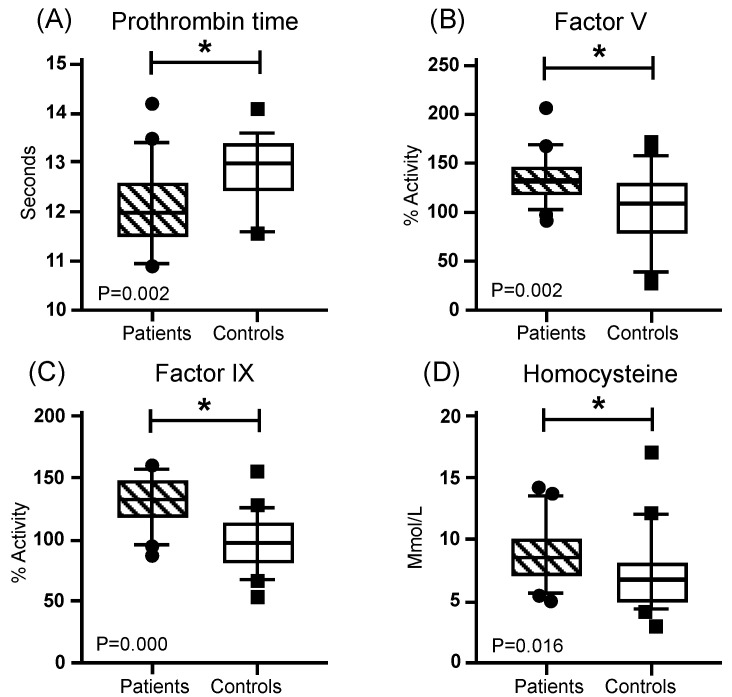
Coagulation comparation. Mustache graphs showing the median as well as the 10–90 percentile and out-of-range values. (•, ■) Outliers are represented with circles outside the mentioned percentiles. The *p*-value comparing both groups (patients and controls) is also included. Stressed with an asterisk (*) are samples whose comparisons presented a significant difference during the comparative analysis when applying the Mann–Whitney U test (* *p* ≤ 0.05). Activity (%) = percentage of activity; μmol/L = micromoles per liter.

**Figure 4 genes-16-01126-f004:**
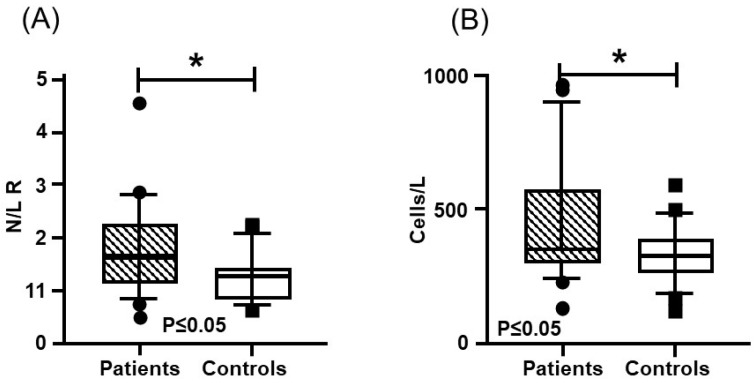
Inflammatory Indexes: (**A**) neutrophil–lymphocyte ratio; (**B**) systemic immune-inflammation index. Mustache graphs showing the median as well as the 10–90 percentile and out-of-range values. (•, ■) Outliers are represented with circles outside the mentioned percentiles. The *p*-value comparing both groups (patients and controls) is also included. Stressed with an asterisk (*) are samples whose comparisons presented a significant difference during the comparative analysis when applying the Mann–Whitney U test (*p* ≤ 0.05).

**Figure 5 genes-16-01126-f005:**
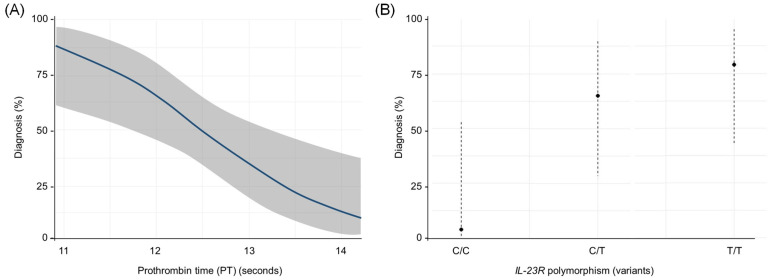
Predicted odds of diagnosis by (**A**) Prothrombin Time and (**B**) *IL-23R* rs1569922 polymorphism. In this model, a lower PT and higher frequency of IL-23R increase the likelihood of LCPD occurrence with high specificity (78.3%) and sensitivity (73.9%) of the cases.

**Figure 6 genes-16-01126-f006:**
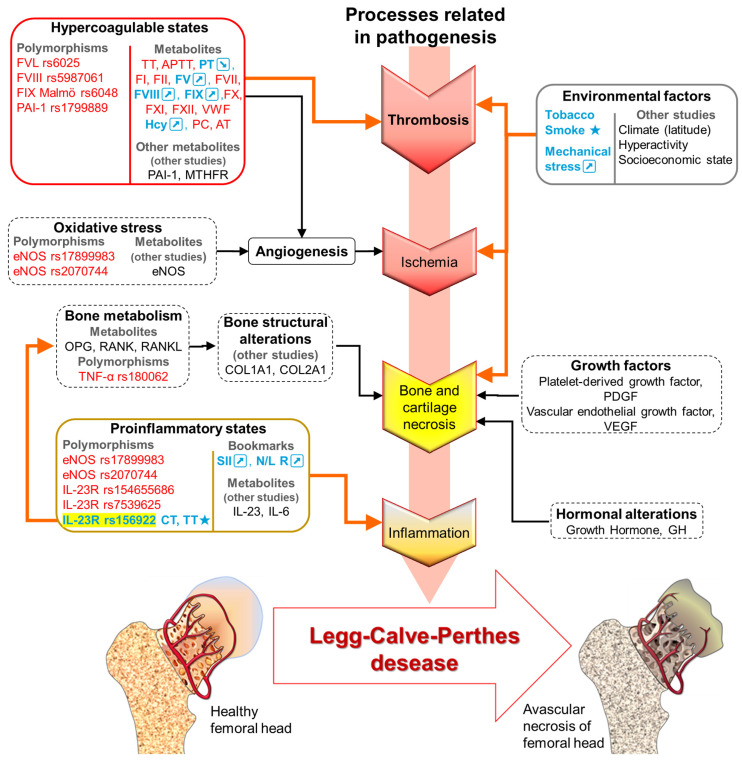
LCPD pathogenesis is complex since multiple mechanisms are affected. Description of results of this study (metabolites and polymorphisms). Different etiological factors will share a common pathogenic pathway that triggers LCPD. The markers not associated with patients with LCPD are shown in red, in blue and/or ★, the associated with patients with LCPD are shown; orange arrows indicate direct alterations present in our population. In black, the markers are described in other studies. Black arrows and dotted lines represent direct alterations described by other authors. TT: Thrombin Time; APTT: Activated Partial Thromboplastin Time; PT: Prothrombin Time; F: coagulation factor: I, II, V, VII, VIII, IX, X, and XI. VWF: Von Willebrand Factor; Hcy: Homocysteine; PC: Protein C; AT: Antithrombin; PAI-1: Plasminogen Activator Inhibitor-1; MTHFR: Methylenetetrahydrofolate reductase; eNOS: Endothelial nitric oxide synthase; OPG: Osteoprotegerin; RANK: Receptor Activator of Nuclear Factor κ B; RANKL: Receptor Activator of Nuclear Factor κB Ligand; TNF-α: Tumor necrosis factor alpha; COL1A1: Collagen type I alpha 1 chain; COL2A1: Collagen type II alpha 1; IL: Interleukin; IL-23R: IL-23 receptor; SSI: Systemic Immunoinflammation Index; N/L R: Neutrophil lymphocyte ratio; ↗: Increases. ↘: Decreases.

**Figure 7 genes-16-01126-f007:**
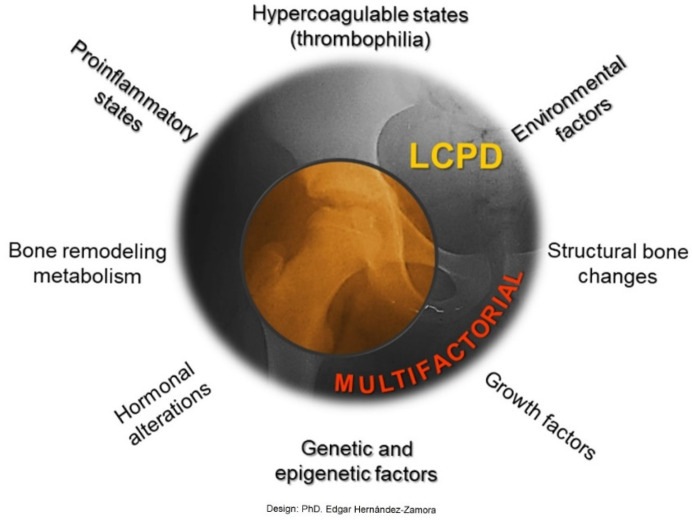
LCPD is a multifactorial disease. Our results indicate risk markers associated with hypercoagulable states (thrombophilia) and proinflammatory states, markers related to bone structural changes and bone remodeling metabolism, as well as some environmental factors.

**Table 1 genes-16-01126-t001:** Partial blood count.

	Patients (N = 23)	Controls (N = 23)	*p*
Hemoglobin	16.2 ± 1.8	15.6 ± 1.7	0.1
Leukocytes	6.2 ± 1.3	6.6 ± 1.6	0.2
Lymphocytes	35.0 ± 9.5	37.5 ± 8.6	0.3
Neutrophils	52.2 ± 11.5	50.8 ± 9.3	0.6
Monocytes	6.0 ± 1.5	6.0 ± 1.3	0.9
Platelets	266.0 ± 56.8	289.3 ± 76.0	0.2

N: Total number of individuals in the study. Variables expressed as mean ± standard deviation, *p* ≤ 0.05.

**Table 2 genes-16-01126-t002:** Coagulation time and factors, antithrombotic proteins, and homocysteine.

Parameter	Patients (N = 23)	Controls (N = 23)	*p*	Effect Size
TT	17.5 ± 1.0	17.2 ± 1.8	0.900	---
PT	**11.9 ± 0.8**	**13.0 ± 0.8**	**0.002 ***	**0.44**
APTT	31.8 ± 2.7	30.1± 3.5	0.200	---
Factor I	356.5 ± 87.0	337.7± 103.4	0.500	---
Factor II	107.8 ± 14.3	119.4 ± 29.4	0.100	---
Factor V	**135.1 ± 26.7**	**103.6 ± 37.5**	**0.002 ***	**0.53**
Factor VII	112.3 ± 22.5	121 ± 32.2	0.200	---
Factor VIII	103.0 ± 24.5	88.3 ± 32.	0.100	---
Factor IX	**130.7 ± 22.0**	**98.6 ± 22.8**	**0.000 ***	**0.20**
Factor X	117.6 ± 18.4	128.6 ± 31.0	0.200	---
Factor XI	108.2 ± 19.4	104.7 ± 44.5	0.700	---
Factor XII	93.7 ± 30.9	91.3 ± 40.5	0.800	---
VWF	87.9 ± 30.1	95.2 ± 41.9	0.500	---
Hcy	**8.9 ± 2.5**	**7.2 ± 3.2**	**0.016 ***	**0.35**
PC	119.8 ± 12.9	110.7 ± 34.7	0.400	---
AT	120.0 ± 16.1	118.4 ± 24.3	0.800	---

N: Total number of individuals in the study. Variables expressed as mean ± standard deviation, parameters in bold indicate statistically significant differences, for which the mean and the standard deviation are presented [X (±SD)]. * *p* ≤ 0.05. TT: Thrombin Time; PT: Prothrombin Time; APTT: activated Partial Thromboplastin Time; VWF: von Willebrand Factor; Hcy: Homocysteine; PC; Protein C; AT: antithrombin.

**Table 3 genes-16-01126-t003:** Systemic Immune-Inflammation Index.

	Patients (N = 23)	Controls (N = 23)	*p* (Effect Size)	Effect Size
N/L R	**1.7 ± 0.8**	**1.1 ± 0.4**	**0.0412 ***	**0.30**
SII	**451.1 ± 232**	**334.8 ± 111**	**0.0360 ***	**0.63**
M/L R	6.2 ± 2.4	6.4 ± 1.9	0.7000	---
P/L R	8.0 ± 2.5	7.8 ± 2.0	0.8000	---

N: Total number of individuals in the study. Variables expressed as mean ± standard deviation, * *p* ≤ 0.05, N/L R: neutrophil-lymphocyte ratio, SII: systemic immune-inflammation index, M/L R: Monocyte-lymphocyte ratio, P/L R: platelet-lymphocyte ratio, parameters in bold text are difference significative.

**Table 4 genes-16-01126-t004:** Association of FVL, FVIII, FIX, PAI-1, eNOS, IL-23R and TNF-α polymorphisms.

SNP and Model	Genotype	Controls n (%)	Patients n (%)	OR (95% CI)	*p*	OR (95% CI) *	*p* *
*FVL* rs6025							
	C/C	23 (100.0%)	22 (100.0%)			1.00	
------------------	C/T	0 (0.0%)	1 (4.3%)	SNP Monomorphic		SNP Monomorphic	0.2 *
	T/T	0 (0.0%)	0 (0.0%)				
*FVIII* rs5987061							
	C/C	23 (100.0%)	23 (100.0%)				
------------------	C/T	0 (0.0%)	0 (0.0%)	SNP Monomorphic		SNP Monomorphic	
	T/T	0 (0.0%)	0 (0.0%)				
*FIX* Malmö rs6048							
	A/A	22 (97.8%)	20 (87.0%)	1.00		1 *	
------------------	A/G	1 (2.2%)	3 (13.0%)	0.3 (0.03–3.1)	0.3	0.2 (0.02–3.1) *	0.2 *
	G/G	0 (0.0%)	0 (0.0%)				
*PAI-1* rs1799889							
	A/A	23 (100.0%)	23 (100.0%)				
------------------	A/G	0 (0.0%)	0 (0.0%)	SNP Monomorphic		SNP Monomorphic	
	G/G	0 (0.0%)	0 (0.0%)				
*eNOS* rs17899983							
	G/G	21 (78.3%)	18 (78.3%)	1.00		1.00 ˄**	
Codominant	G/T	1 (4.3%)	4 (17.4%)	0.2 (0.02–2.0)		0.2 (0.02–2.0) ˄**	
	T/T	1 (4.3%)	1 (4.3%)	0.9 (0.05–14.7)	0.3	7.43 (0.04-NA) ˄**	0.2 **
Dominant	G/G	21 (91.3%)	18 (78.3%)	1.00		1.00 ˄**	
	G/T-T/T	2 (8.7%)	5 (21.7%)	0.3 (0.06–2.0)	0.2	0.4 (0.06–2.7) ˄**	0.3 **
Recessive	G/G-G/T	22 (95.7%)	22 (95.7%)	1.00		1.00 ˄**	
	T/T	1 (4.3%)	1 (4.3%)	1.0 (0.06–17.0)	1.0	7.8 (0.02–0) ˄**	0.8 **
*eNOS* rs2070744							
	T/T	18 (78.3%)	17 (73.9%)	1.00		1.00 **	
Codominant	C/T	4(17.4%)	6 (26.1%)	0.7 (0.2–2.6)	0.4	0.6 (0.12–3.1) **	0.4 **
	C/C	1 (4.3%)	0 (0.0%)				
Dominant	T/T	18 (78.3%)	17 (73.9%)	1.00		1.00 **	
	C/T-C/C	5 (21.7%)	6 (26.1%)	0.8 (0.2–3.06)	0.7	0.54 (0.09–3.3) **	0.5 **
Recessive	T/T-C/T	22 (95.7%)	23 (100.0%)	1.00	2.0	1.00 **	0.3 **
	C/C	1 (4.3%)	0 (0.0%)				
*IL-23R* rs1569922							
	T/T	6 (26.1%)	11 (47.8%)	**1.00**		**1.00 ****	
Codominant	C/T	10 (43.5%)	11 (47.8%)	**1.7 (0.3–6.2)**	**0.03**	**1.7 (0.4–7.8) ****	**≤0.05 ****
	C/C	7 (30.4.2%)	1 (4.3%)	**12.9 (1.3–13.05)**		**27.0 (1.9–398) ****	
Dominant	T/T	6 (26.1%)	11 (47.8%)	1.00		1.00 **	
	C/T-C/C	17 (73.9%)	12 (52.2%)	2.6 (0.8–8.9)	0.1	2.9 (0.8–11.2) **	0.1 **
Recessive	T/T-C/T	16 (69.6%)	22 (95.7%)	**1.00**		**1.00 ****	
	C/C	7 (30.4%)	1 (4.3%)	**12.83 (1.3–86.2)**	**0.01**	**19.05 (1.60–226.44) ****	**≤0.05 ****
*IL-23R* rs7518660							
	G/G	19 (82.6%)	15 (65.2%)	1.00		1.00 ˄**	
------------------	G/A	0 (0.0%)	0 (0.0%)	0.4 (0.1–1.5)	0.2	0.5 (0.1–1.9) ˄**	0.2 ˄**
	A/A	14 (17.4%)	8 (34.8%)				
*IL-23R* rs7539625							
	G/G	16 (69.6%)	18 (78.3%)	1.00		1.00 **	
Codominant	G/A	7 (30.4%)	4 (17.4%)	2.0 (0.5–8.0)	0.3	2.2 (0.5–9.6) **	0.3 **
	A/A	0 (0.0%)	1 (4.3%)				
Dominant	G/G	16 (69.9%)	18 (78.3%)	1.00		1.00 **	
	G/A-A/A	7 (30.4%)	5 (21.7%)	1.6 (0.4–6.0)	0.5	1.6 (0.4–6.6) **	0.5 **
Recessive	G/G-G/A	23 (100%)	22 (95.7%)	1.00	0.2	1.00 **	
	A/A	0 (0.0%)	1 (4.3%)				0.2 **
*TNF-α* rs7518660							
	G/G	22 (95.7%)	22 (95.7%)	1.00		1.0 **	
------------------	G/A	0 (0.0%)	0 (0.0%)	1.00 (0.06–17.0)	1.0	8.6 (0.07–0.0) **	0.3 **
	A/A	1 (4.3%)	1 (4.3%)				

SNP: Single nucleotide polymorphism. rs: Reference SNP. n: sample size. A: Adenine. T: Thymine. G: Guanine. C: Cytosine. OR: Odds ratio. CI: confidence interval. ˄: No Hardy–Weinberg equilibrium. *: adjusted using PT, FV, FIX, Hcy. **: Adjusted using SSI and RN/L. *FVL*: Factor V Leiden. *FVIII*: Factor eight. *FIX*: Factor nine. *eNOS*: endothelial nitric oxide synthase. *TNF-α*: tumor necrosis factor alpha. *IL23R*: interleukin 23 receptor. Bold: increased risk of developing LCPD.

## Data Availability

The datasets used in this study are available from the corresponding author upon reasonable request.

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
