# Peer review of "Association of *IL-23R* rs1569922 and Other Probable Frequent Etiological Factors with Legg–Calvé–Perthes Disease in Mexican Patients"

_genes, 2025, doi:10.3390/genes16101126_

Round 1

Reviewer 1 Report

Comments and Suggestions for Authors

Title and abstract

The title is clear and informative, specifying both the genetic factor (IL23R rs1569922) and the disease (Legg Calve Perthes disease in Mexican patients).

The abstract provides a good overview. The inclusion of many polymorphisms in the abstract makes it harder to follow the main findings. Highlighting IL-23R as the central novel association more prominently is recommended.

Introduction

The introduction is comprehensive with a well detailed background on LCPD. The reason for investigating both hemostatic and inflammatory polymorphisms is good. The link between IL-23R polymorphisms and bone metabolism is introduced but could be better framed as a knowledge gap leading to the study/ or as a secondary aim.

Methods

The design (case-control) is appropriate for a rare disease. The inclusion of 23 cases and 23 controls is a limitation due to the small sample size. Matching by age, sex, and BMI is a strength.

Laboratory protocols are well described, though the methods section is at times overly technical (listing commercial kit catalog numbers is not necessary for a high-impact journal).

Statistical methods are appropriate, including chi-square, Mann–Whitney U, Hardy Weinberg equilibrium, and logistic regression. no correction for multiple testing was mentioned, which is a serious limitation given the number of polymorphisms analyzed.

Ethical considerations are adequately addressed.

Results

The results are detailed and supported with tables and figures.

Findings on PT, Factor V, Factor IX, homocysteine, and IL-23R are clearly reported.

narrative is heavy with numerical detail, which could be improved.

The logistic regression model is well explained

The inclusion of environmental factors (passive smoking, sports activity) is interesting, though these data are not deeply analyzed statistically.

Discussion

The discussion is extensive and connects findings with previous literature.

The role of IL-23R in inflammatory bone disease is argued correctly.

The discussion sometimes overstates the implications given the small cohort.

The claim that IL-23R rs1569922 polymorphism is a “risk factor” for LCPD should be tempered with acknowledgment.

Strengths include linking genetic, inflammatory, and environmental risk factors.

Weaknesses include limited focus too many elements dilute the main genetic finding. The limitations are acknowledged

References

The reference list is extensive and covers both classical and recent studies.

Author Response

Introduction

  • The introduction is comprehensive with a well detailed background on LCPD. The reason for investigating both hemostatic and inflammatory polymorphisms is good. The link between IL-23R polymorphisms and bone metabolism is introduced but could be better framed as a knowledge gap leading to the study/ or as a secondary aim.

Answer: In the introduction, we highlighted that the role of some polymorphisms related to the IL23R gene has been associated with osteonecrosis, but they have been very poorly studied. Thank you.

Methods

  • Laboratory protocols are well described, though the methods section is at times overly technical (listing commercial kit catalog numbers is not necessary for a high-impact journal).

Answer: The catalog numbers have been retired. Thank you.

  • No correction for multiple testing was mentioned, which is a serious limitation given the number of polymorphisms analyzed.

Answer: We did not perform multivariate analysis for the comparison between groups due to the sample size. Therefore, a univariate analysis was performed. Regarding polymorphisms, these were adjusted using SSI and RN/L. Thank you.

Results

  • Narrative is heavy with numerical detail, which could be improved.

Answer: In several sections, the IL-23R polymorphism RS was removed. Thank you.

  • The inclusion of environmental factors (passive smoking, sports activity) is interesting, though these data are not deeply analyzed statistically.

Answer: We felt it was pertinent to report that our population presented several environmental factors, primarily secondhand smoke. However, this was not the objective of analyzing them, as these variables were not established in the controls. These were only data reported by the patients. Thank you.

Discussion

  • The discussion sometimes overstates the implications given the small cohort. The claim that IL-23R rs1569922 polymorphism is a “risk factor” for LCPD should be tempered with acknowledgment.

Answer: We replaced "risk factor" with "factor associated with this study population." Thank you.

  • Weaknesses include limited focus too many elements dilute the main genetic finding. The limitations are acknowledged.

Answer: At the end of the discussion and in the conclusion, greater emphasis was placed on the genetic finding.

References

  • The reference list is extensive and covers both classical and recent studies.

Answer: We've reduced the number of references (from 53 to 49), thank you.

We appreciate your comments. It's a pleasure to respond.

Reviewer 2 Report

Comments and Suggestions for Authors

The article entitled Association of IL-23R rs1569922 and Other Probable Frequent Etiological Factors with Legg-Calvé-Perthes Disease in Mexican Patients presents valuable research on the role of genetic factors (with particular emphasis on IL-23R rs1569922 polymorphism), haemostatic and inflammatory factors in the pathogenesis of Legg-Calvé-Perthes disease (LCPD). The paper is well structured, includes a reliable description of methods, and provides a detailed statistical analysis. The results indicating a significant association between IL-23R polymorphism and the risk of LCPD constitute a new contribution to the literature on the subject and may open up directions for further research. The paper requires several corrections before further processing and acceptance for publication. Detailed comments are provided below.

Major comments:

  1. The small sample size (23 patients and 23 controls) limits the statistical power and the possibility of generalizing the results. I recommend emphasizing this limitation more clearly and suggesting the need for studies on larger and multicenter populations to confirm the results obtained.
  2. The lack of a validation group or replication in other populations means that the results may be specific only to the studied cohort. The need for replication in independent groups to confirm the association of IL-23R polymorphism with LCPD should be clearly indicated.
  3. The limited discussion of the translational significance of biomarkers means that the practical implications remain unclear. It would be worthwhile to expand the discussion on the potential use of biomarkers (e.g., IL-23R rs1569922, PT, Hcy) in the diagnosis or monitoring of disease progression.
  4. The introduction is too extensive in places and contains repetitions, which obscures the main objective of the study. The authors could shorten this section and focus more on the existing data on IL-23R and its role in musculoskeletal diseases, together with the latest references, e.g. DOI: 10.3390/app15126896 ; https://doi.org/10.3390/life12111799; https://doi.org/10.1016/j.joca.2022.09.005 ;
  5. The lack of detailed data on the control group, especially in terms of environmental factors (e.g., exposure to tobacco smoke, physical activity), may affect comparisons. I recommend supplementing this information to strengthen the credibility of the analysis.
  6. The final conclusions are quite general – it would be worthwhile to expand on them, emphasizing the practical implications and potential of using IL-23R rs1569922 as a biomarker for LCPD risk. This could be supplemented by a simplified graphic diagram summarizing the main pathogenetic pathways.

Author Response

  • The small sample size (23 patients and 23 controls) limits the statistical power and the possibility of generalizing the results. I recommend emphasizing this limitation more clearly and suggesting the need for studies on larger and multicenter populations to confirm the results obtained.

Answer: We've included this comment at the end of the discussion. Thank you.

  • The lack of a validation group or replication in other populations means that the results may be specific only to the studied cohort. The need for replication in independent groups to confirm the association of IL-23R polymorphism with LCPD should be clearly indicated.

Answer. We've included this comment at the end of the discussion. Thank you.

  • The limited discussion of the translational significance of biomarkers means that the practical implications remain unclear. It would be worthwhile to expand the discussion on the potential use of biomarkers (e.g., IL-23R rs1569922, PT, Hcy) in the diagnosis or monitoring of disease progression.

Answer. We've included this comment in the conclusion. Thank you.

  • The introduction is too extensive in places and contains repetitions, which obscures the main objective of the study. The authors could shorten this section and focus more on the existing data on IL-23R and its role in musculoskeletal diseases, together with the latest references, e.g. DOI: 10.3390/app15126896 ; https://doi.org/10.3390/life12111799; https://doi.org/10.1016/j.joca.2022.09.005 ;

Answer: We shortened the introduction a bit and emphasized some points related to IL-23R. Thank you.

  • The lack of detailed data on the control group, especially in terms of environmental factors (e.g., exposure to tobacco smoke, physical activity), may affect comparisons. I recommend supplementing this information to strengthen the credibility of the analysis.

Answer: We included a sentence clarifying that this was not the primary objective of the study and only described the presence of some environmental factors present in our patients. Thank you.

  • The final conclusions are quite general – it would be worthwhile to expand on them, emphasizing the practical implications and potential of using IL-23R rs1569922 as a biomarker for LCPD risk.

Answer: We include this comment in the conclusion. Thank you.

  • This could be supplemented by a simplified graphic diagram summarizing the main pathogenetic pathways.

Answer: We consider that figures 6 and 7 describe in detail and finally in a summarized manner the pathogenic pathways.

We appreciate your comments. It's a pleasure to respond.